# When Unstructured Big Text Corpus Meets Text Network Analysis: Social Reality Conceptualization and Visualization Graph of Big Interview Data of Heavy Drug Addicts of Skid Row

**DOI:** 10.3390/healthcare11172439

**Published:** 2023-08-31

**Authors:** Israel Fisseha Feyissa, Nan Zhang

**Affiliations:** 1School of Global Studies, Kyungsung University, Busan 48434, Republic of Korea; 2Global Migration Research Center, Kyungsung University, Busan 48434, Republic of Korea; 3Department of Social Welfare, Jeonbuk National University, Jeonju 54896, Republic of Korea; sishou1122@hotmail.com

**Keywords:** text network graph analysis, unstructured interview data, big data, data-driven approach, topical clusters, health informatics

## Abstract

Relying on user-generated content narrating individual experiences and personalized contextualization of location-specific realities, this study introduced a novel methodological approach and analysis tool that can aid health informatics in understanding the social reality of people with a substance-use disorder in Skid Row, Los Angeles. The study also highlighted analysis possibilities for big unstructured interview text corpus using InfraNodus, a text network analysis tool. InfraNodus, which is a text graph analysis tool, identifies pathways for meaning circulation within unstructured interview data and has the potential to classify topical clusters and generate contextualized analysis results for big narrative textual datasets. Using InfraNodus, we analyzed a 1,103,528-word unstructured interview transcript from 315 interview sessions with people with a substance-use disorder, who narrated their respective social realities. Challenging the overgeneralization of onlookers, the conceptualization process identified topical clusters and pathways for meaning circulation within the narrative data, generating unbiased contextualized meaning for the collective social reality. Our endeavors in this research, along with our methodological setting and selection, might contribute to the methodological efforts of health informatics or the conceptualization and visualization needs of any big text corpus.

## 1. Introduction

Skid Row, an area approximately 50 square blocks in size and situated east of the Financial District and Historic Downtown Center of Los Angeles, has been a hub for homeless-related services since the 1970s and is the primary concentration of homelessness. It is estimated that between 12,000 and 14,000 individuals live in this area [1]. Illicit drugs are widely prevalent among Skid Row residents, and they are commonplace within many homeless communities [2,3]. While discussing the social realities of Skid Row’s residents, there is a risk of only focusing on pathology due to the associated stigma it can create [4]. However, it is also undeniable that illicit drugs are widespread within many homeless communities [2,3]. Therefore, there is a need to provide contextualized data and analysis that can inform public health departments, incorporate perspectives from the community at risk [5], and be practical enough to develop more sensitive interventions [6].

An ongoing challenge for researchers and health officials addressing healthcare issues in locations like Skid Row is the lack of stable data associated with its residents, such as aspects related to daily activity spaces, patterns of mobility, social and professional networks, and even the contact “home” address [7]. One successful aid for healthcare efforts in areas like Skid Row, in the form of eHealth, is Spatial video geo-narratives (SVG) [8,9]. SVG offers an innovative interview method that incorporates spatially encoded video recording alongside commentary. This technique enables the mapping and analysis of significant spaces and their associated meanings or interpretations of social realities. SVG has proven successful in documenting drug and marginalized environments in various places throughout the United States, with particular emphasis on its application in Skid Row, Los Angeles [8,9]. Although SVG allowed a environmentally cued narrative where the place is used to stimulate discussion about fine-scale geographic characteristics of an area and its context, our study aimed to focus on individual narrations to contextualize the common experiences or social realities of residents of Skid Row while maintaining its characteristics as a location for prevalent substance abuse and destitution. As mentioned in Curtis and colleagues’ study [10], there is a necessity of including individual narratives alongside with the result of SVG to fully understand what was happening in a specific location. To this effect, we relied on user-generated content narrating individual experiences and personalized contextualization of location-specific realities that can potentially inform healthcare initiatives.

A substantial volume of user-generated content disseminated on social media platforms has given rise to a plethora of enduring, publicly accessible, and easily retrievable fragments pertaining to individuals’ private lives. Consequently, investigative endeavors within numerous domains of the social sciences have increasingly turned to the wealth of content data available on these online platforms, as they provide a platform for individuals to express their thoughts, experiences, and beliefs in an unfiltered and authentic manner [11]. Moreover, this phenomenon has generated an extensive corpus of big data that are readily accessible for analysis. Nonetheless, the availability of such data necessitates the utilization of appropriate methodologies for extracting, organizing, and deriving meaningful insights from it.

The conventional methodological inquiries and analysis approaches employed in the realm of social sciences are currently being confronted by the abundance and diversified nature of data accessible through social media platforms. In varying degrees, traditional methodologies such as ethnographical, statistical, and computational techniques face significant challenges in adequately capturing the multitude of ways in which social media data can contribute to our comprehension of online users’ behaviors [12]. Additionally, issues related to representativeness and sampling biases further complicate efforts to investigate social behavior within the context of social media platforms [11].

Ethnographic methodologies utilized in social data analysis, particularly in the context of social media, have often focused on qualitative-ethnographic approaches [13,14,15,16]. These investigations often focus much on qualitative-ethnographic approaches, which are less effective, time-consuming, and often irrelevant when applied to big data sets. On the other hand, traditional statistical techniques employed in social data analysis typically involve the use of case data matrices, with the social media user or content serving as the unit of analysis. For instance, statistical approaches employed in the analysis of YouTube videos aim to comprehend users’ social behavior within communities by examining discussions (comments) surrounding a particular video and conducting sentiment analysis [17,18,19]. These techniques necessitate manual coding by the researcher, the implementation of rigorous sampling techniques, and the use of appropriate text analysis software. Despite their relevance, the application of traditional statistical approaches to big data sets is not common due to the laborious and complex nature of organizing and coding such extensive datasets.

Computational methods have been extensively tested and proven to be suitable for collecting social data, offering potential insights of interest to social scientists. Studies argue that a computational methodology of topic modeling is a useful method for the descriptive analysis of unstructured social media data sets and is best used as part of a mixed-method strategy, with topic model results guiding deeper qualitative analysis [20]. However, traditional approaches to social data analysis, such as content analysis, have limitations when it comes to providing reliable analysis results from large-scale data sets, primarily due to the time-consuming nature of the process [12]. Unfortunately, for most social scientists, achieving compelling results using computational approaches often requires proficiency in computer science skills, including manipulation via Application Programming Interface (API). These computational approaches present a challenge for social scientists, as they must navigate the acquisition of substantial or enormous metadata sets while adhering to the limitations imposed by a web-services approach. Moreover, computational approaches that involve API manipulation are inherently focused on investigating the inner workings of social media platforms rather than users’ social behavior. Recently, the introduction of GPT-3 (Generative Pre-trained Transformer) [21] has revolutionized big data analysis due to its capacity to process large text corpora and perform tasks like text generation, sentiment analysis, and Natural Language Processing, facilitating insights extraction from extensive textual data.

This study aimed to explore a possible analysis approach for big social data to provide an unbiased interpretation of social behavior or conceptualization of a phenomenon within an unstructured big interview data set. The article presents a text network analysis approach that contextualizes the experience of addiction and destitution within an interview transcript of 288 individual storytellers. The interview data collectively constitute a large text corpus of 1,103,528 words. The study investigated the social realities of people with a substance-use disorder (type of drug not specified), people with a substance-use disorder, people with a substance-use disorder, fentanyl addicts, and heroin addicts separately. While introducing our approach to dealing with such large unstructured interview data, this study aimed to conceptualize the common experience of mental health problems, homelessness, addiction, and destitution.

Methodologically, through a data-driven approach, this study aimed to reach an organized clusters of topics that can help to conceptualize of the social realities of the individuals interviewed. As is common in the research practice of qualitative methods like thematic analysis, the conceptualization of social realities of individuals who live at the fringe of any society often relies on the observations and definitions of the researcher [22,23]. Such interpretations are often met with interpretive bias. The aim of the study is, thus, to employ the computational possibilities of InfraNodus software [24] to guide our investigation of extracting the exact conceptualization of individual/common experience, without the inclusion of the authors’ inference as to what the definitions might be. 

Using InfraNodus, our study employed a Text network graph analysis. Text network graph analysis is intended to extract important words and topical clusters within the narrations of the individuals interviewed. Using the concept graph created by the text network analysis, we extracted and inferred analytical meaning from the connections between words and important topic clusters. This study is likely to inform innovative and novel methodological approaches, and analysis tool that can aid researchers who are dealing with big unstructured interview data or large text corpus. Our study especially highlighted procedures related to data management, data cleaning, data analysis, bias control, generalizability, interpretation, and tips for time efficiency. We believe our study holds significant implications for the field of thematic analysis, particularly when applied to unstructured big interview data. Our innovative approach combines the challenges of handling unstructured interview data with a data-driven methodology, enriched by the utilization of the InfraNodus analysis tool. This unique synergy allows for a systematic, objective, and efficient exploration of themes and patterns within the data, overcoming the limitations of traditional thematic analysis methods. By integrating these elements, our research not only contributes to advancing our understanding of the thematic analysis process but also offers a practical solution for researchers grappling with the complexities of unstructured big interview data analysis, thus paving the way for more robust and insightful research outcomes. 

## 2. Materials and Methods

### 2.1. Justification for Using Youtube Videos for Analysis

The vast and diverse content available on YouTube has been recognized by many researchers as a valuable resource for understanding the world from the perspective of video uploaders [25,26,27]. However, there is no consensus among researchers on the ethical use and evaluation of social media data [12,28]. Some researchers obtain permission from uploaders [29,30,31], while others address ethical issues and justify their use of such data [32,33,34]. The main justification for using YouTube videos is that they are in the public domain as soon as they are uploaded, and uploaders understand that anyone with internet access can access their videos. Similar to some previous studies, the authors of this study faced uncertainties when using YouTube videos [35,36] but were justified their use based on the content provided and the information analyzed. Informed consent was deemed to be given for the use of the contents, and efforts were made to ensure confidentiality. The decision to use the data was also justified because of the unique research opportunity it presented, as people with a substance-use disorder are a vulnerable group whose social realities are often hidden. Personal information was deliberately avoided in the analysis to protect privacy.

### 2.2. Sample

This study utilized a sample of 288 individual cases collected in the form of life story narration video documents. All individual cases were obtained from Mark Lalita’s YouTube channel, “Soft White Underbelly”, where he interviewed hundreds of heavily destitute individuals living in the Skid Row neighborhood in downtown Los Angeles. These interviews were available in the public domain and were used for our analysis.

The interviewer, Mark Lalita, had a unique position in understanding the reality of his interviewees. Although he had no predefined frameworks and questions to structure the inquiry, throughout his work, there was a questioning pattern that gave room to the storyteller to structure their story and their own social reality [37] Much of Mark’s approach when interacting with his subjects illuminated a raw and honest depiction of their respective realities. Since the start of this channel, he has conducted many interviews of residents on Skid Row. His project has helped share the stories of prostitutes, people with a substance-use disorder, pimps, strippers, sexual abuse survivors, incest survivors, homeless individuals, and many more at the fringe of society. His channel has allowed for the understanding of individuals in some of the lowest times of their lives to be seen [38].

The sample extracted from this channel featured individuals with heavy drug addiction. Initially, our sample included a total of 700 video clips. These video clips were selected from the long playlist of the channel according to the titles given by the content creator. The titles indicated the name of the person interviewed and the type of drug they are abusing, for example, “Crystal Meth Addict Interview—Demon”. Video clips of second-time interviews or updates on interviewees were also included in the sample after careful assessment of their relevance to the overall analysis.

After selecting a total of 522 video clips using the titles given, relevancy assessment, duplicate exclusion, and application of inclusion and exclusion criteria, this study finally prepared 315 videos for analysis.

The relaxed atmosphere of the interview sessions allowed for a pure depiction of the social realities of the interviewees. The interviewer was not interrogative; rather, he gave room for the interviewees to structure their own stories. Using YouTube videos for text network analysis in this fashion enabled the identification of subjective words and statements [39]. Often, studies on the experiences of individuals are skewed to the interpretation of researchers. However, the interview sessions in which the interviewees structured their narration allowed for unbiased inference of their social realities. Here, using text network analysis to extract the interviewees’ exact depictions and inferences of their social realities allowed for deeper analysis, decreasing the chance of subjective interpretation bias [39].

As an unbiased view of the experiences of the subjects at hand, our sample and method of collection had their own unique features. First, it saved the difficult and time-consuming task of locating and recruiting people with a substance-use disorder who might not otherwise follow through with the strict procedures of a typical experimental research project. Second, although the availability of the interview sessions in a YouTube format also made the computational part of the analysis easily manageable, the relevancy assessment and criteria for the inclusion and exclusion of samples are described in detail in the following subsections.

### 2.3. Relevancy Assessment

The relevancy of the videos for text network analysis was assessed by the basic information given on the video content as to who the person is interviewed, the type of drug used, and the clarity of conversation flow with the interviewer. The authenticity of the content creator, Mark Lalita, was also assessed by checking the consistency of the content on the channel as well as the legitimacy of the uploader as a content creator on YouTube. In addition, the relevant videos were videos in which the interviewees explicitly defined and provided information about their respective social realities. 

First initial selection of potential videos and their verification for relevance was conducted. The initially selected videos were then verified for relevancy by the primary research investigator. These selected videos were filtered, collected, and organized for analysis. The collection of interview transcripts from the finally selected videos was also performed in the same fashion.

### 2.4. Inclusion and Exclusion Criteria

After the videos were selected and after their transcripts were collected, the content of the videos was segregated based on the inclusion and exclusion criteria stated on Table 1.

### 2.5. Analysis Strategy: Concept Generating Using Text Network Analysis and Data Analysis

The initial step in data analysis involved gathering captions and transcribing the chosen videos. Within the videos, the interviewees narrated their story in their own fashion. Here, for the analysis tool to decode the flow, the task of collecting the captions and later transcribing them required maintaining the original form of how the dialogue took place. 

InfraNodus, a tool developed by Nodus Labs in Leeds, UK, served as an insight-generating platform designed for text network analysis. This tool employs an algorithm to identify pathways for the circulation of meaning within textual data. It achieves this by visualizing transcribed text as a graph and extracting essential metrics for both individual concepts and the text as a whole using network analysis techniques.

InfraNodus is instrumental in transforming disjointed textual data into a coherent structure. It accomplishes this by representing the text corpus as a network, revealing key topics, their interrelationships, and structural gaps. Moreover, it enables a comprehensive analysis of discourse structure and the assessment of its diversity based on the graph’s community structure.

Utilizing the resulting transcribed text data and graph representation, we employed InfraNodus to identify crucial topical clusters that describe distinct social realities. This process also highlighted influential words associated with these clusters. The text network analysis algorithm organized the transcribed data into a network and pinpointed the most influential words within the discourse through word co-occurrence analysis. Additionally, the graph community detection algorithm identified various topical clusters, shedding light on the primary themes in the text and their interconnections.

To provide meaningful interpretations of the identified topical clusters and associated words, we employed thematic analysis to gain insights into the interviewees’ social realities. Throughout this thematic analysis, we meticulously examined the influential words and topical clusters to better understand their precise representations and interpretations within the context of the interviewees’ experiences and social worlds. Thematic interpretation played a crucial role in elucidating the contextual meaning inferences of words that appeared to hold influence during the analysis, necessitating a return to the raw data for a more comprehensive understanding.

### 2.6. Text Network Analysis with Using InfraNodus Tool

By employing the InfraNodus tool for analysis, the researcher’s responsibility was to input the raw textual corpus into the tool precisely as it was originally formatted. Throughout this process, the researchers exercised caution to ensure that the content of the transcribed data remained unaltered, thus preventing any potential distortion of the conceptualization process. Therefore, before feeding the raw textual corpus organizing, cleaning, and adjusting in separate categories of analysis groups were performed outside of InfraNodus. 

Right after the organized, cleaned, and categorically separated text corpus is inserted into InfraNodus tool, the tool recommends important step workflow suggestions for better data visualization. This process involves several key steps for analyzing and visualizing text data. Initially, text is added or imported. Subsequently, the text is transformed into an interactive network graph, revealing essential patterns and relationships. Users then explore this visualization, identifying influential concepts and topical clusters, and can selectively hide irrelevant words. GPT-3 AI generates high-level ideas, unifying concepts in clusters. Relevant and top concepts are pinpointed for analysis, along with the exploration of key relations. The main topical clusters, formed by recurring concept patterns, are examined. Insights are recorded, helping comprehension and retention. Users can zoom in on specific topics, locate discourse entry points, and reveal underlying ideas by removing influential concepts. The process is reiterated for a deeper understanding, and the graph can be reset when needed. Finally, analysis insights are summarized and shared, contributing to a comprehensive understanding of the text data.

In simpler terms, the tool initiates the process by eliminating syntax and converting words into their morphemes, thereby reducing redundancy. For instance, “neighborhoods” is transformed into “neighborhood”. Subsequently, the tool proceeds to eliminate articles, conjunctions, auxiliary verbs, and select other stop words like “he”, “she”, or “the”. Following this, the remaining transformed words are depicted as nodes in a network graph, with edges denoting their co-occurrences. The tool emphasizes nodes (words) that frequently appear together and identifies clusters of nodes (words) that exhibit a higher tendency to co-occur. Ultimately, the analysis yields a visual network graph representation of the text. Through the use of colors and sizes, the tool provides a clear visualization of the text data’s structure and the prevalent topics within it. Then, the research inquiry continues with exploring topical clusters and what the tool generated as insights.

### 2.7. Data Sample

The data type, individual count, and the special characteristics of the sample used for this study are shown in Table 2 The study captured the 288 individual narrations in 315 videos that produced a corpus text of 1,103,528 words. 

## 3. Results

### 3.1. Establishing Concept Graphs

This study employed a data-driven approach using InfraNodus software to analyze the social realities of individuals interviewed. This study sought to leverage the computational capabilities of InfraNodus software to guide the investigation and extract an objective conceptualization of individual and shared experiences, while avoiding the inclusion of the authors’ inferences regarding the definitions. To address this, we generated six concept graphs corresponding to each data sample indicated in Table 2.

#### 3.1.1. Drug Addicts

The concept graph under the case identifier of people with a substance-use disorder revealed organized clusters of topics related to drug addiction and its various aspects. As it is shown in Figure 1 and Figure 2, one significant cluster was the “Opioid Crisis”, comprising 19% of the nodes, which included keywords such as heroin, meth, rehab, and fentanyl. The “Drug Network” cluster, accounting for 18% of the nodes, explored connections, regrets, and encounters with drug-related experiences. The “Sobriety Journey” cluster (15%) delved into the challenges and milestones of maintaining sobriety. “Family Dynamics” (12%) examined the role of family relationships in the context of addiction, while “Money Struggles” (11%) focused on financial hardships. Additionally, clusters such as “Youth Culture” (11%), “Urban Life” (10%), “Skid Row Living” (1%), and “Hotel Living” (1%) shed light on various aspects of living conditions and experiences associated with addiction.

#### 3.1.2. Life on Drugs

This concept graph highlighted various aspects of life on drugs, resulting in the generation of ten distinct topical clusters. These topical clusters provided insights into different dimensions of drug-related life experiences. As it is shown in Figure 3 and Figure 4, the “Family Moving” cluster (16%) explores the impact of familial relocation on individuals’ lives, encompassing elements such as geographic changes, parental roles, and personal memories. The “Drug Metering” cluster (13%) explored the initiation and regulation of drug consumption, including factors like dosage, substances used, and temporal considerations. Additionally, the “Fort Lauderdale” cluster (13%) delved into the influence of location on drug-related experiences, including financial aspects and reflections on one’s youth. The “Recovery Treatment” cluster (13%) shed light on aspects related to seeking and undergoing treatment for addiction, including residential facilities and the transformative nature of the recovery process. Other clusters, such as “Addiction Kick” (13%), “Behavior Drawing” (9%), “Baltimore Story” (8%), “Street Selling” (6%), “High Schooling” (4%), and “Time Cutting” (4%), further elucidated various dimensions of drug-related experiences and their consequences.

#### 3.1.3. Crack Addicts

The conceptual graph of the “crack addicts” sample produced ten distinct topical clusters that shed light on various aspects of experiences related to crack addiction (see Figure 5 and Figure 6). The “Family Dynamics” cluster (20%) explored the complexities of familial relationships, encompassing emotions such as love, childhood memories, and the impact of family dynamics on individuals’ lives. The “Street Life” cluster (19%) delved into the challenges and realities of being homeless, including experiences of homelessness, survival, and encounters with danger. The “Homelessness” cluster (16%) highlighted the struggles faced by people with a substance-use disorder in securing stable housing and the hardships associated with being homeless. Moreover, the “Family Ties” cluster (12%) examined the connections and interactions within families affected by addiction, while the “Life Changes” cluster (10%) delved into the transformative experience’s individuals underwent during their journey towards sobriety. The “Substance Abuse” cluster (10%) focused on the patterns and consequences of crack addiction, including the age of onset, drug use, and the challenges of recovery. Additionally, clusters such as “Job Struggles” (7%) shed light on the difficulties faced in maintaining employment, while “Self-Care” (3%) emphasized the importance of personal wellbeing. The clusters “Skid Row” (1%) and “High School” (1%) provided insights into specific locations and educational experiences relevant to the story tellers.

#### 3.1.4. Crystal Meth Addicts

The crystal meth addicts sample resulted in the identification of nine distinct topical clusters (see Figure 7 and Figure 8). The “Time Living” cluster (20%) delved into the passage of time, encompassing aspects such as life on the streets, periods of sobriety, and the challenges faced while incarcerated. The “Parenting” cluster (20%) shed light on the complexities of being a parent, including the impact of one’s own upbringing, drug use, and family dynamics on parenting practices. The “Emotions” cluster (11%) explored the range of feelings experienced by the addicts, such as love, strength, and moments of vulnerability. Moreover, the “Reality Struggles/Profanity” cluster (11%), expressed in words of profanity, examined the harsh realities they encountered, financial struggles, and challenging circumstances. The “Addiction Care” cluster (9%) emphasized the importance of community support and addressing addiction-related issues. Additionally, clusters such as “Homelessness Beat” (8%), “Family Loss” (7%), “Life Changes” (7%), and “Trauma Recovery” (5%) provided insights into the unique challenges faced by the people with a substance-use disorder, including experiences of homelessness, family tragedies, life-altering changes, and the process of recovering from traumatic events.

#### 3.1.5. Fentanyl Addicts

The Fentanyl addicts’ sample produced 11 distinct topical clusters. When examining the topical clusters of graphs related to fentanyl addicts, several key themes emerged (see Figure 9 and Figure 10). The largest cluster, comprising 14% of the nodes, focused on time management and the challenges faced by addicts in organizing their lives. Words like time, make, work, and prison indicated the difficulties they encounter in maintaining a structured routine. The second cluster, accounting for 13% of nodes, revolved around drug abuse and the progression from heroin to fentanyl addiction. Keywords such as start, addiction, problem, and homeless highlighted the severe consequences of substance abuse. Parental loss was another significant cluster at 13%, exploring the emotional impact of the death of a parent or close friend on addicts. Words like die, afraid, remember, and thinking reflected the profound sense of loss and grief experienced. Family dynamics, accounting for 12% of nodes, delved into complex relationships and issues of abandonment or abuse involving parents, siblings, and other family members. Urban living, also at 12%, focused on the challenges of life on the streets, including financial struggles and the need to survive. Youth struggles made up 10% of the nodes, highlighting the difficulties faced by young addicts in finding stability, employment, and healthy relationships. Mental health, self-reflection, re-entry after incarceration, self-awareness, and high schooling each formed smaller clusters ranging from 2% to 7% of nodes, offering insights into the specific challenges within these areas.

#### 3.1.6. Heroin Addicts

The heroin addicts’ sample produced nine distinct topical clusters (see Figure 11 and Figure 12). When analyzing the topical clusters of graphs related to heroin addicts, several key themes emerged. The most significant cluster, accounting for 18% of the nodes, revolved around financial struggles. Words like money, job, hard, homeless, and making reflected the challenges addicts faced in managing their finances and securing stable employment. The second cluster, also at 18%, focused on substance abuse, particularly the journey from initial drug use to becoming addicted to heroin. Keywords like start, addiction, stop, and fentanyl indicated the progression of substance abuse and the detrimental consequences it brings, including homelessness. Incarceration formed a notable cluster at 15%, shedding light on the experiences of addicts in the criminal justice system. Keywords such as jail, prison, hope, and change suggested a desire for transformation and a reflection on past mistakes. The recovery process, comprising 12% of nodes, explored the path towards sobriety, emphasizing the importance of supportive relationships and rehabilitation programs. Parenting struggles accounted for 11% of nodes, delving into the challenges faced by addicts in raising their children while battling addiction. Mental health, at 10%, addressed the emotional wellbeing of addicts, with keywords like feel, bad, and problem indicating the presence of mental health issues. Sibling rivalry, homelessness, and adolescence form smaller clusters ranging from 5% to 6% of nodes, highlighting specific challenges within these areas. These clusters provided a comprehensive overview of the multifaceted issues faced by heroin addicts, encompassing financial hardships, substance abuse, incarceration, recovery, parenting struggles, mental health concerns, sibling dynamics, homelessness, and the unique challenges of adolescence.

### 3.2. Influential Discourse Elements within Concept Graphs

Applying the analysis steps from 5 to 14 of Table 2, influential discourse elements within each graph were highlighted. Applying these steps mainly relied on the assumptions of betweenness centrality [40]. Betweenness centrality is a method for identifying the level of impact a node (keyword) holds in regulating the movement of information within a graph. Its primary application is to locate nodes that act as connectors between different sections of the graph. Nodes with high betweenness centrality were deemed influential in the concept graph since they govern the flow of information between other nodes. The influential elements within the discourse of the narrations were with relatively higher betweenness centrality. As was mentioned in step no. 9, in the conceptual graph, some concepts driven by influential keywords had high influence even if they did not appear often. These were the easiest entry points into this discourse. Thus, locating discourse entrance points is highly important for a coherent thematic analysis. In addition, as mentioned in step no. 14, the analysis also requires removing the most influential concepts from the graph to reveal the underlying ideas around them. This step will help extract the nuance and explore the peripheral topics that make a discourse special and can potentially serve as low-barrier entrance points.

The discourse on drug addiction (conceptual graph 1) encompassed various topical clusters, each with its own set of influential keywords. The “Opioid Crisis” cluster, accounting for 19%, discussed the issues regarding the opioid crisis through influential keywords like heroin, meth, fentanyl, and rehab. These keywords in this topical cluster informed the importance of discussing the addiction experience from the perspective of opioid crisis and from its impact on people with a substance-use disorder. The Drug Network cluster, at 18%, emphasized the significance of understanding social connections and network dynamics through influential keywords such as people, friend, sell, and addiction. The Sobriety Journey cluster (15%) acknowledged the challenging path towards sobriety with keywords like sober, hard, stop, and worked, emphasizing the need for support. Family Dynamics (12%) underscored the impact of familial relationships on drug addiction, with keywords like family, brother, homeless, and jail. The Money Struggles cluster (11%) highlighted the influence of financial instability on people with a substance-use disorder, using keywords like money, work, and crack. The Youth Culture cluster (11%) explored the role of youth culture in drug addiction through keywords like school, high, and weed. Urban Life (10%) delved into the specific challenges faced by individuals in urban environments with keywords like street, life, and childhood. The Skid Row Living (1%) and Hotel Living (1%) clusters touched upon the experiences of people with a substance-use disorder in specific living conditions associated with homelessness and transience. While their percentages were relatively low, these clusters still acknowledged the influence of such living situations on the experiences people with a substance-use disorder.

Conceptual graph 2, Life on Drug, presents diverse topical clusters contributing to the discourse surrounding drug addiction experiences. The “Family Moving” cluster (16%) examined the impact of relocation on individuals, with influential keywords like ended, left, Florida, and moving. These keywords underscore the need to understand how moving affect the addiction experience. The “Drug Metering” cluster (13%) focused on the initial stages of drug use, employing keywords such as start, give, month, and fentanyl. This cluster touched upon the patterns and factors associated with the beginning of drug addiction. “Fort Lauderdale” (13%) explored the influence of location, highlighting keywords like place, money, and Lauderdale. These keywords underscored the need to consider the specific characteristics of certain locations in comprehending drug addiction experiences (Fort Lauderdale in this case). The “Recovery Treatment” cluster (13%) centered on the process of recovery, referencing keywords like point, days, center, and treatment. This cluster highlighted the importance of discussing the various aspects and approaches to addiction recovery. The “Addiction Kick” cluster (13%) emphasized the journey of overcoming addiction, utilizing keywords like people, wanted, and kick. This cluster touched upon the challenges and milestones related to addiction recovery. The “Behavior Drawing” cluster (9%) explored personal development and self-expression, mentioning keywords such as artist, behavior, and drawing. These keywords illuminated the influence of creative outlets and personal growth in the discourse surrounding drug addiction. The “Baltimore Story” cluster (8%) highlighted the experiences and challenges faced in Baltimore, employing keywords like clean, working, and Baltimore. These keywords further underscored the significance of location-specific narratives in understanding drug addiction. The “Street Selling” cluster (6%) addressed the illicit drug market, mentioning keywords like drug, sell, home, and street. These keywords emphasized street-level drug transactions in the discourse. The “High Schooling” (4%) and “Time Cutting” (4%) clusters explored the impact of education and time management, respectively, with keywords like school, high, and time. While with lower percentages, these clusters shed light on specific aspects of drug addiction experiences.

Conceptual graph 3 presents a comprehensive overview of the context of people with a substance-use disorder’ context, offering insights into the discourse surrounding their experiences. The “Family Dynamics” cluster (20%) emphasized the importance of family relationships, mentioning keywords like child, love, mother, and father. These keywords underscored how family dynamics impact the lives of people with a substance-use disorder. The “Street Life” cluster (19%) shed light on the challenges and experiences associated with being homeless, using influential keywords like street, living, and sleep. The “Homelessness” cluster (16%) addressed the hardships faced by people with a substance-use disorder due to homelessness, employing influential keywords like money, hard, and homeless. These keywords highlighted the significance of discussing the impact of homelessness on the lives of people with a substance-use disorder. The “Family Ties” cluster (12%) explored the connections within families, utilizing keywords like mom, sister, and dad. This cluster exposed the role and type of family ties in crack addiction experiences. The “Life Changes” cluster (10%) reflected the transformative nature of crack addiction, referencing keywords like year, clean, and prison. This cluster signified the changes and challenges individuals face throughout their crack addiction journey. The “Substance Abuse” cluster (10%) focused on the patterns and characteristics of crack addiction, mentioning keywords like drug, crack, and addiction. This cluster emphasized more on substance abuse in the discourse surrounding people with a substance-use disorder. The “Job Struggles” cluster (7%) highlighted the difficulties faced in maintaining employment, utilizing keywords like job, lost, and reason. These keywords pointed to the challenges people with a substance-use disorder encounter in the job market. With a relatively low percentage within the discourse, the “Self-Care” cluster (3%) underscored self-care practices. Similarly, the “Skid Row” (1%) and “High School” (1%) clusters acknowledged specific living conditions and educational contexts, respectively, with keywords like Skid Row and high school. Although they received less attention in the discourse, these clusters still shed light on specific location-specific aspects of crack addiction experiences.

The topical clusters associated with people with a substance-use disorder in conceptual graph 4 offer valuable insights into the discourse surrounding crystal meth addiction experience. The “Time Living” cluster (20%) emphasized the temporal aspects of crystal meth addiction, passage of time, and the challenges faced by the homeless, using keywords like time, street, and sober. The “Parenting” cluster (20%) shed light on the challenges faced by people with a substance-use disorder in their roles as parents, mentioning keywords like mom, dad, leave, left, and family. These influential keywords highlighted the significance of understanding the impact of crystal meth addiction on parenting experiences. The “Emotions” cluster (11%) explored the emotional experiences of people with a substance-use disorder, the emotional aspects of addiction and recovery, utilizing keywords like feel, love, and sick. The “Reality Struggles/Profanity” cluster (11%) focused on the challenges and profanity-laden realities, employing keywords like fu*k, sh*t, and money. The “Addiction Care” cluster (9%) highlighted the importance of addressing addiction and the necessary support for recovery, mentioning keywords like community and care. The “Homelessness Beat” cluster (8%) addressed the experiences of homelessness in relation to crystal meth addiction, using keywords like school and homeless. The “Family Loss” cluster (7%) explored the impact of family loss on people with a substance-use disorder and the profound effects of family loss in the context of crystal meth addiction, employing keywords like daughter and show. The “Life Changes” cluster (7%) reflected the transformative nature of crystal meth addiction experiences, changes, and challenges individuals face throughout their crystal meth addiction journey, mentioning keywords like drug, change, and hard. The “Trauma Recovery” cluster (5%) focused on the process of recovering from trauma, utilizing keywords like rape and broke. Although with a lower percentage, this cluster highlighted experiences of trauma recovery in the discourse surrounding crystal meth addiction.

The topical clusters associated with fentanyl addicts (conceptual graph 5) offer insights into the discourse surrounding their experiences. The “Time Management” cluster (14%) highlighted the need for discussing time-related challenges in the context of fentanyl addiction and time spent in a difficult circumstance, using keywords like time, work, long, hour, prison, and happen. The “Drug Abuse” cluster (13%) focused on the abuse of drugs, mentioning keywords like drug, addiction, and heroin. The “Parental Loss” cluster (13%) addressed the loss of parents and the profound impact of parental loss on individuals struggling with fentanyl addiction, using keywords like dad, friend, and die. The “Family Dynamics” cluster (12%) explored the dynamics within families, mentioning keywords like mom, left, and family. The influential keywords emphasized the influence of family relationships on fentanyl addiction experiences. The “Urban Living” cluster (12%) shed light on the challenges and aspects of living in urban environments, using keywords like life, street, and living. The “Youth Struggles” cluster (10%) highlighted the struggles faced by young individuals, using keywords like good, job, and love. The influential keywords illuminated the unique challenges of fentanyl addiction. The “Mental Health” cluster (7%) focused on the intersection of mental health and addiction, mentioning keywords like sick and pain. The influential keywords signified the importance of considering the mental health aspect within the discourse surrounding fentanyl addiction. The “Self-Reflection” cluster (6%) emphasized self-reflection and introspection, using keywords like kind and feel. The “Re-Entry” cluster (6%) addressed the process of re-entering society after periods of incarceration, mentioning keywords like back and clean. The influential keywords signified the importance of supporting individuals in their transition from incarceration to recovery. The “Self-Awareness” cluster (6%) highlighted self-awareness and personal growth, using keywords like know and day. The “High Schooling” cluster (2%) acknowledged the educational context of high school, using keywords like high and school. Although with a lower percentage, this cluster underscores the potential impact of high school experiences on fentanyl addiction.

The topical clusters associated with heroin addicts (conceptual graph 6) provided insights into the discourse surrounding their experiences. The “Financial Struggles” cluster (18%) emphasized the challenges of managing finances, using keywords like money, job, and homeless. The “Substance Abuse” cluster (18%) focused on the abuse of substances, issues, and consequences associated with substance abuse, mentioning keywords like drug, heroin, and addiction. The “Incarceration” cluster (15%) shed light on the experiences of incarceration, using keywords like jail, prison, and change. This percentage highlighted the impact of incarceration on the lives of heroin addicts and the challenges they faced during and after incarceration. The “Recovery Process” cluster (12%) addressed the journey of recovery, mentioning keywords like rehab, friend, and believe. These influential keywords emphasized the process and support systems involved in recovering from heroin addiction. The “Parenting Struggles” cluster (11%) explored the challenges faced by heroin addicts in their roles as parents, using keywords like family, love, and childhood. These influential keywords highlighted the importance of understanding the impact of heroin addiction on parenting experiences. The “Mental Health” cluster (10%) focused on the influence of mental health, using keywords like feel and bad. The “Sibling Rivalry” cluster (6%) addressed the dynamics between siblings, mentioning keywords like brother and sister. Although with a lower percentage, this cluster highlighted the influence of sibling relationships on the experiences of heroin addiction. The “Homelessness” cluster (5%) shed light on the experiences of homelessness, using keywords like street and tent. Similarly, the “Adolescence” cluster (5%) acknowledged the experiences of heroin addicts during adolescence, mentioning keywords like age and girl. Despite its minor focus, this cluster highlighted the importance of considering the unique challenges faced by adolescent heroin addicts.

Based on the similar themes mentioned within each conceptual graph, Table 3 highlights the similar patterns among the six conceptual graphs. The patterns among these conceptual graphs indicate different aspects of drug addiction and its related challenges. Each graph focuses on a specific theme or aspect, such as the opioid crisis, family dynamics, homelessness, or mental health. These patterns highlight the diverse range of experiences and issues associated with different types of drug addiction. The evolution of themes throughout the whole corpus (all six samples) is also mentioned in this table. The evolution of themes shows how the main topics and the most influential keywords evolved throughout the corpus times the number of concept occurrence. The tone difference within the evolution of these should be read alongside with the order of the organized phrases for the collective conceptualization of the whole corpus and individually for evolution within the concept graphs.

## 4. Discussion

This study aimed to explore one analysis possibility in big social data that can be applied for unbiased interpretation of social behavior or conceptualization of a phenomenon within an unstructured big interview data set. By employing a data-driven approach, this study successfully identified and organized clusters of topics, offering valuable insights into the social realities experienced by individuals with various types of drug addiction in Skid Row. Furthermore, this research examined the computational capabilities of InfraNodus software, specifically its capacity to facilitate the conceptualization of individual and collective experiences, without incorporating the authors’ subjective inferences and assumptions regarding the definitions involved.

The analysis within the conceptual graphs proved instrumental in contextualizing the experience of addiction across individuals with different types of addiction. By identifying common themes across the various types of drug addiction, our investigation was able to delineate the scope of shared experiences. Moreover, the conceptualization process not only provided a coherent organization of the storytellers’ narratives but also offered an efficient and unbiased interpretation of their social realities. Overall, the result of the study and the whole experience of working within the steps of analysis in InfraNodus showed us the potential of the software in how it potentially aids or replace methodological concerns of thematic analysis when dealing with big unstructured interview data. Thematic analysis provides researchers with a transtheoretical tool or technique rather than a comprehensive research methodology grounded in theory. While it offers substantial flexibility in terms of theoretical perspectives and design choices, researchers must exercise meticulous conceptual and design considerations to ensure the methodological integrity [41]. In addition, recent developments in thematic analysis also recommend that researchers apply reflexive thematic analysis [42]. Reflexive thematic analysis acknowledges the influence of the researcher’s own perspectives, assumptions, and experiences on the analysis process [42,43]. It recognizes that researchers are active participants in the research process and that their interpretations are subjective and shaped by their own backgrounds and biases and ask researchers to engage critically with their own interpretations and biases, while still providing a systematic and rigorous approach to analyzing qualitative data. For such concern, InfraNodus allowed a full data-driven text network-analysis approach that will keep the researcher’s bias away from the conceptualization process.

One can also imagine the near impossibility or difficulty of applying rigorous thematic analysis to a big, unstructured interview text corpus. Rather than attempting thematic analysis, applying text network analysis to big and unstructured interview data provides researchers with a powerful tool to explore and understand the relationships and connections between different concepts, themes, and ideas that might otherwise be too difficult or time-consuming. Additionally, text network analysis allows researchers to uncover hidden patterns and relationships that may not be apparent through traditional qualitative analysis methods. However, what AI-based text network analysis might lack in analyzing qualitative data, such as our own, are the following: contextual understanding and nuances of context and cultural subtleties of the circumstances of our storytellers; lack of emotional intelligence with regard to understanding the emotions, intentions, and motivations of our storytellers; and a failure to capture the richness of subjective human experiences. Thus, our study informed researchers who choose to apply text network analysis to have reservations about “studying things in their natural settings, attempting to make sense of, or interpret, phenomena in terms of the meanings people bring to them” [44]. Our study allowed the data to speak for itself by deliberately skipping steps 10 to 13 of the analysis workflow. However, working with those steps will enhance the results by adding the human element to the analysis, which primarily relies on patterns and statistical correlations in the data.

Overall, weighing the potential of text network analysis for qualitative inquiry, it brought to us the possibilities of visualizing text data in a network graph, with words (nodes) representing concepts or themes and edges representing the relationships between them. Unlike other qualitative inquiry methods, this approach allowed us to systematically visualize the connections between different concepts and identify clusters of related topics. Within the scope of the result of this study, we can confidently infer or conceptualize the social realities of the individuals’ interviewees to revolve around the wide variety themes mentioned in Table 3.

As investigators of the sampled data, we can conclude that the stories portrayed in the conceptual graphs had elements of the opioid crisis, substance abuse, and drug networks, emphasizing the prevalence and impact of drugs in their respective society. The stories also included the journey towards sobriety, recovery treatment, and addiction care, shedding light on the process of overcoming addiction and seeking support. Topical clusters on family dynamics, parenting struggles, and sibling rivalry, alongside experiences related to financial struggles and money-related challenges, also explored the complexities of familial relationships within the context of addiction. The themes of youth culture, high schooling, and youth struggles focused on the unique experiences and challenges faced by young individuals dealing with addiction. Urban life and urban living highlighted the specific difficulties and circumstances associated with addiction in urban environments. Homelessness and its related challenges were explored in the family moving and homelessness beat clusters. The cluster of drug metering, location-specific addiction experience, street selling, street life, and Skid Row shed light on the intricate dynamics of drug use and distribution in specific locations. Themes like behavior drawing, self-care, self-reflection, and self-awareness emphasized personal growth and introspection within the context of addiction. Time cutting, time living, and time management reflected the temporal aspects and challenges individuals face in addiction. Emotions, reality struggles/profanity, trauma recovery, and mental health delved into the emotional experiences, struggles with reality, and the need for trauma recovery and mental well-being within addiction. Family loss and life changes highlighted the transformative and often tragic events that occur within families affected by addiction. The themes of re-entry and incarceration explored the challenges faced by individuals as they navigate the criminal justice system. Finally, adolescence focused on the specific issues faced by young people grappling with addiction during this developmental stage. These organized phrases provided a comprehensive overview of the multifaceted themes and issues related to drug addiction, highlighting the diverse range of experiences and challenges within this context.

## 5. Conclusions

InfraNodus and the workflow chart used to develop the conceptual graphs in this study could serve an important purpose for qualitative analysis. In addition, our endeavor in this research and our methodological selection might contribute to the efforts of health informatics and the analysis of massive data in the health field through techniques related to artificial intelligence which are helping to create new procedures and applications. Firstly, the tool we used helped us and will help researchers identify key themes and concepts by mapping out the relationships between different elements in unstructured big interview data. This allows for a clearer understanding of the most significant themes present. Secondly, the concept graphs generated enabled us to explore the connections and relationships between different themes, uncovering the underlying patterns and structures within the data exposing untampered social reality depictions of the study participants. Thirdly, by examining the centrality of nodes in the network, the graphs helped us identify key players and influencers in the interview data, shedding light on words that hold significant positions or exert influence within the themes being studied. Additionally, removing the most influential concepts allowed for a deeper exploration of peripheral topics, enabling a more nuanced understanding of the underlying ideas, additionally filtering the exact depictions of life in Skid Row. Lastly, by comparing network graphs of different interviewees, we can discern differences and similarities in the way individuals conceptualize and connect different themes and concepts, contributing to a deeper understanding of the variations within the dataset. Overall, the conceptual graphs we generated in this study and the workflow can serve as valuable tools in qualitative analysis, facilitating the identification of themes, exploration of connections, analysis of influence, and comparison of conceptualizations within different data set.

Reflecting on the findings from the analysis, the conceptual graphs produced a story that incorporated interconnectedness of various aspects related to drug addiction, including social factors, personal experiences, mental health, family dynamics, and the challenges faced by individuals in different contexts. From a practice point of view, these insights can inform researchers, policymakers, and practitioners working in the field of addiction by providing a data-driven and location specific understanding of the multifaceted nature of drug addiction and its implications.

Overall, as per the aim of this study, the application of InfraNodus software and the analysis of concept graphs offer a powerful computational tool to explore and extract objective conceptualizations of individual and shared experiences related to drug addiction. This approach complements traditional qualitative analysis methods by providing a data-driven perspective and uncovering hidden patterns and relationships that might not be apparent through traditional means.

## 6. Limitations of the Study

Although the capability of the tool in our research design was remarkable and we are satisfied that it met our needs for a data-driven analysis approach, we still feel the need to mention how much expert reasoning in the analysis process is important for achieving better comprehensive results. One of the strengths of high-quality qualitative analysis lies in the emphasis on the analyst’s lived experience [45] and how that experience can shape the interpretive framework of the study. In this matter, we, the investigators, did not possess the lived experience of being addicted to drugs to the extent of our informants or have direct experience working with individuals struggling with drug addiction. Importantly, our analysis approach should have incorporated the expertise and positionality of the human analyst if these factors are explicitly acknowledged. While topical clusters are estimated mathematically, their meaning is derived qualitatively, necessitating a process of meaning-making that is inseparable from the analyst.

## Figures and Tables

**Figure 1 healthcare-11-02439-f001:**
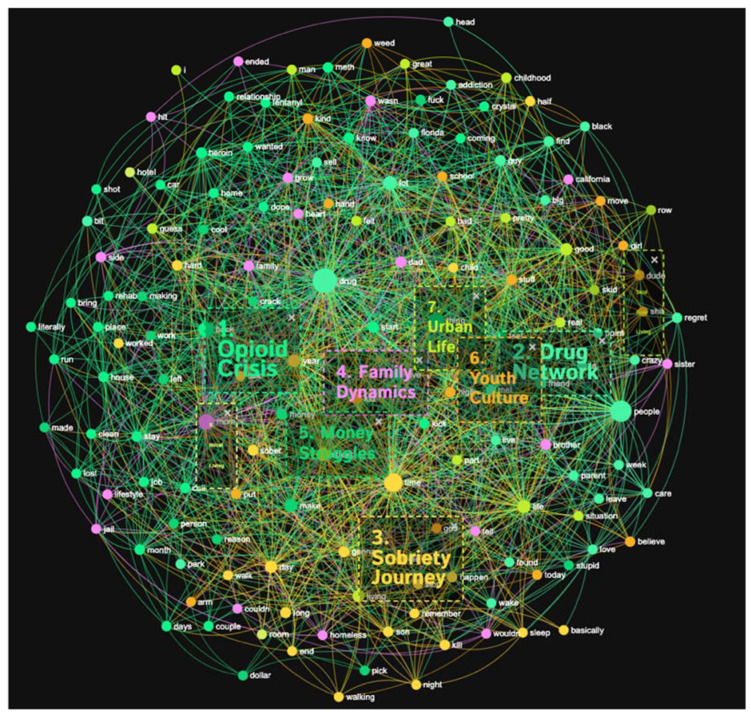
Concept graph 1 “Drug addicts” sample.

**Figure 2 healthcare-11-02439-f002:**
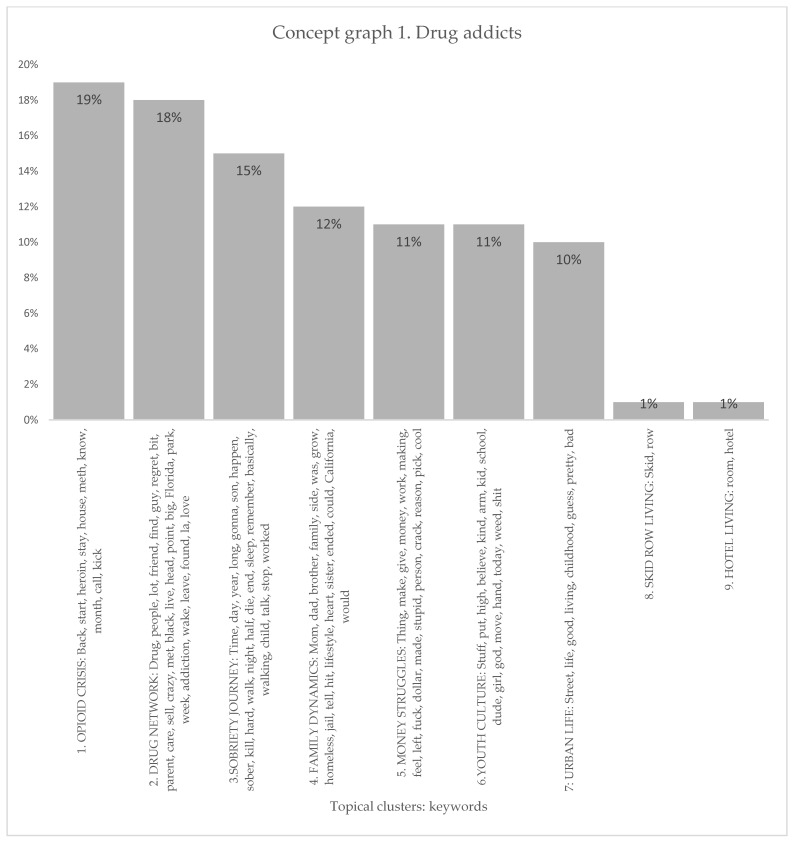
Topical clusters in concept graph 1 “Drug addicts” sample.

**Figure 3 healthcare-11-02439-f003:**
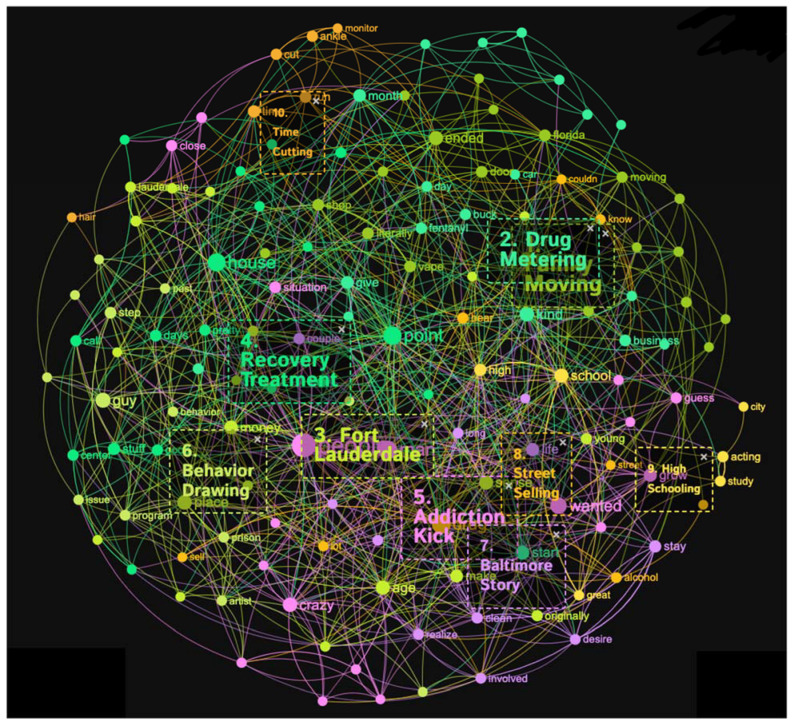
Concept graph 2 “Life on drug” sample.

**Figure 4 healthcare-11-02439-f004:**
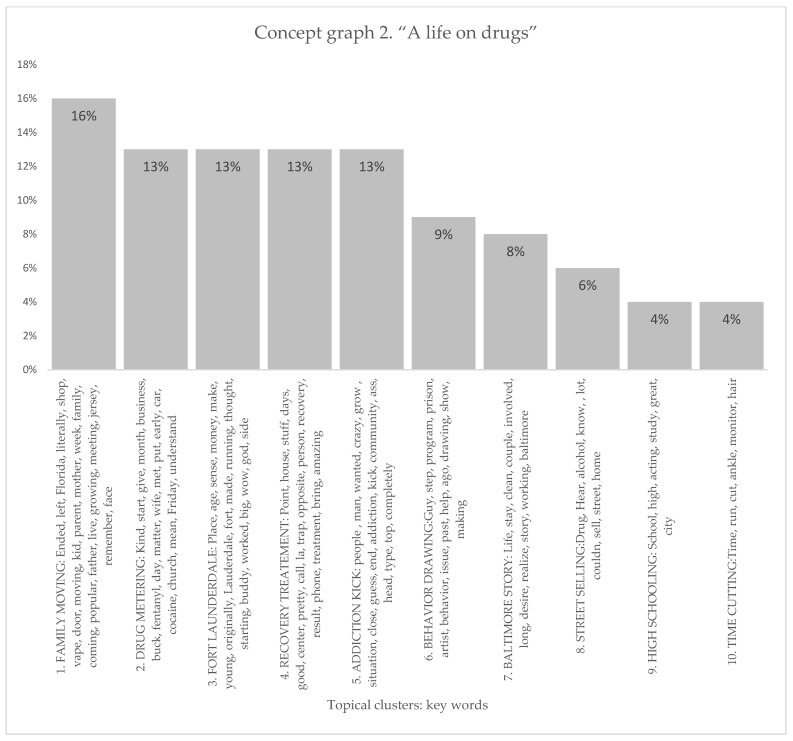
Topical clusters in concept graph 2 “Life on drug” sample.

**Figure 5 healthcare-11-02439-f005:**
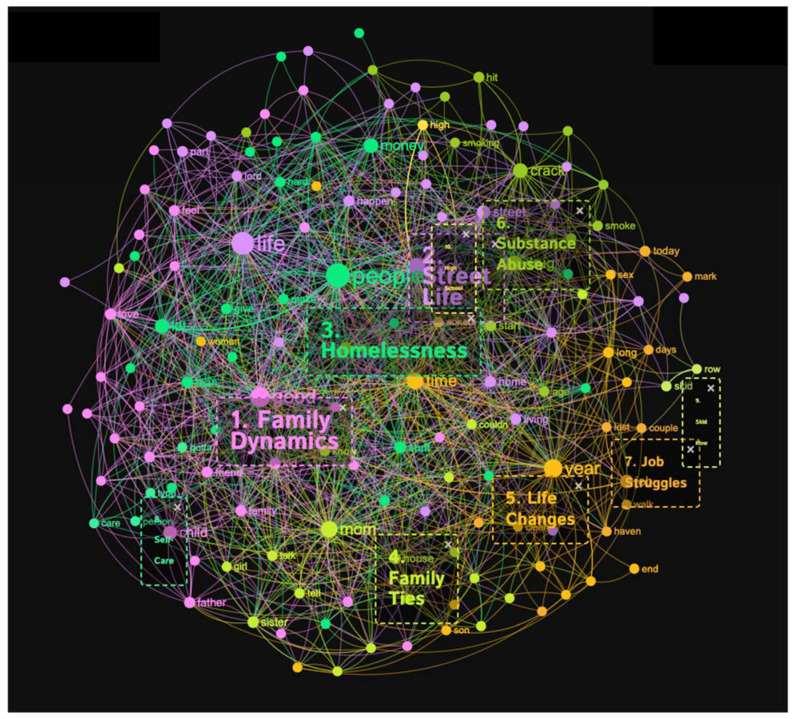
Concept graph 3 “Crack addicts” sample.

**Figure 6 healthcare-11-02439-f006:**
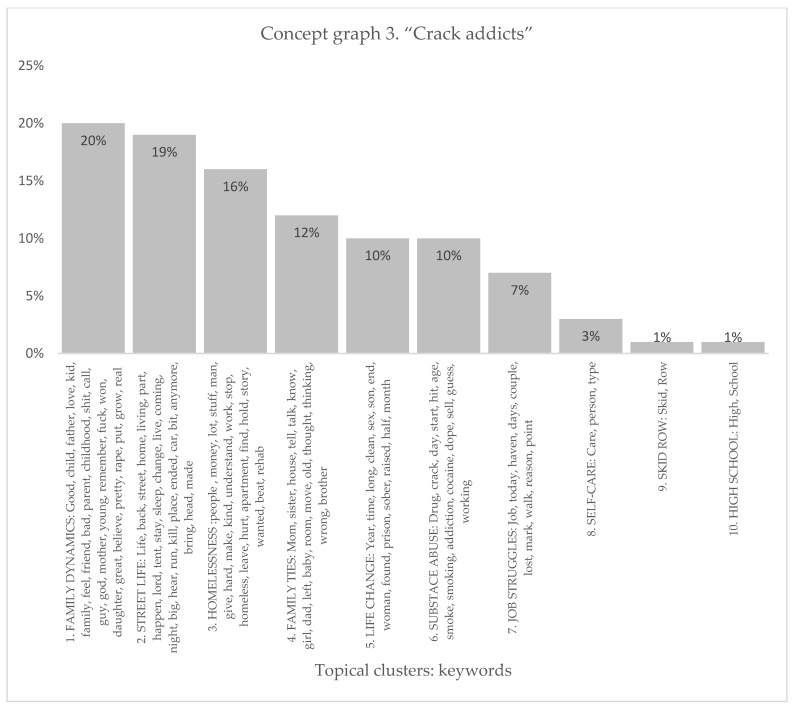
Topical clusters in concept graph 3 “Crack addicts” sample.

**Figure 7 healthcare-11-02439-f007:**
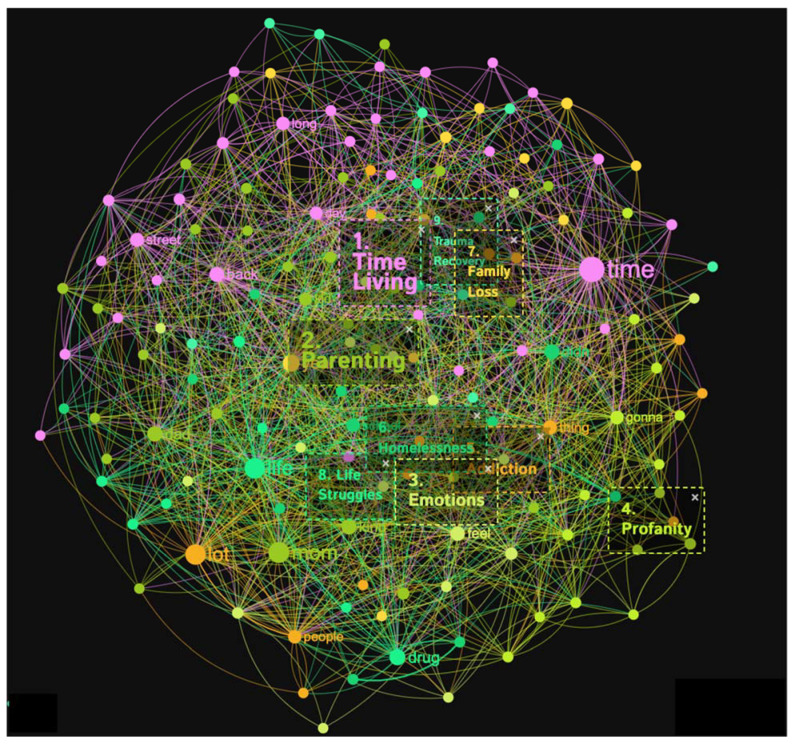
Concept graph 4 “Crystal meth addicts” sample.

**Figure 8 healthcare-11-02439-f008:**
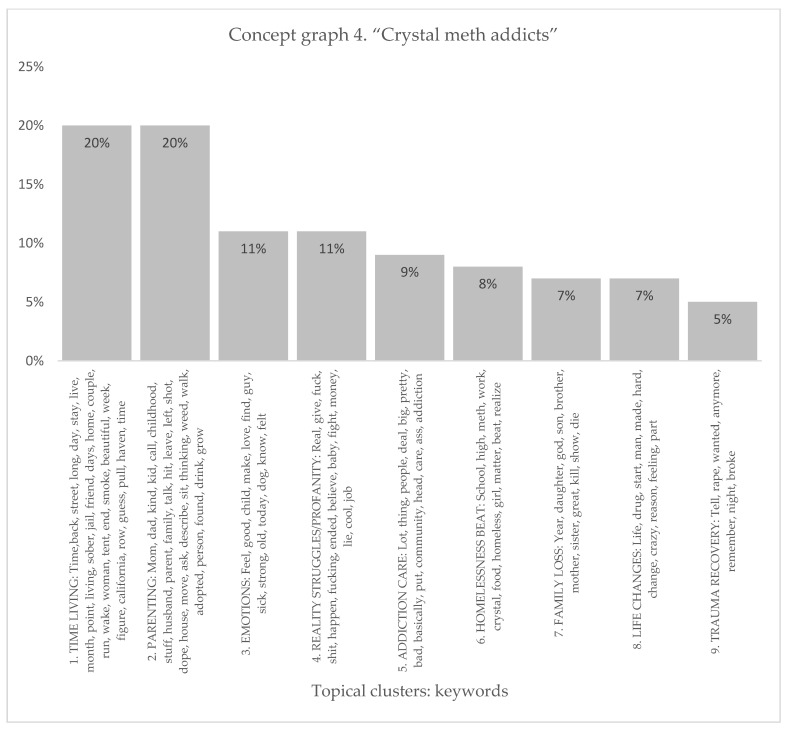
Topical clusters in concept graph 4 “Crystal meth addicts” sample.

**Figure 9 healthcare-11-02439-f009:**
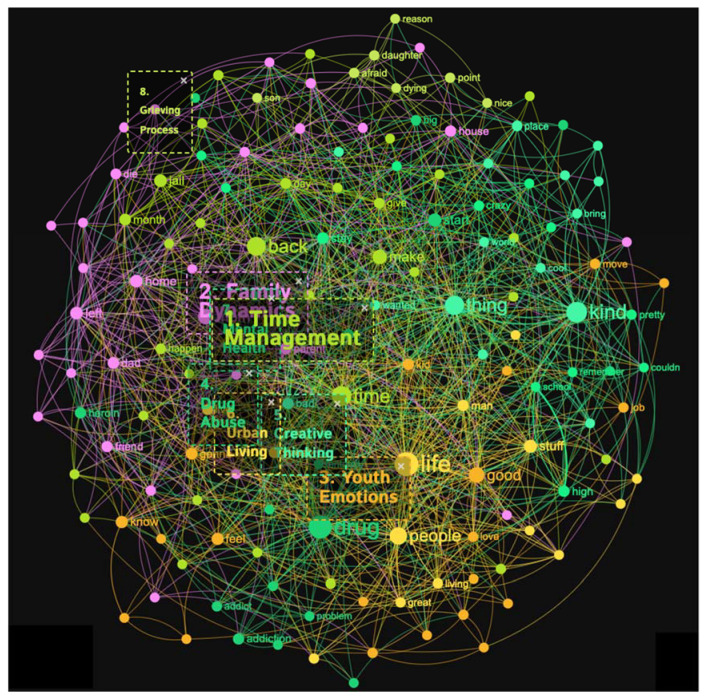
Concept graph 5 “Fentanyl addicts” sample.

**Figure 10 healthcare-11-02439-f010:**
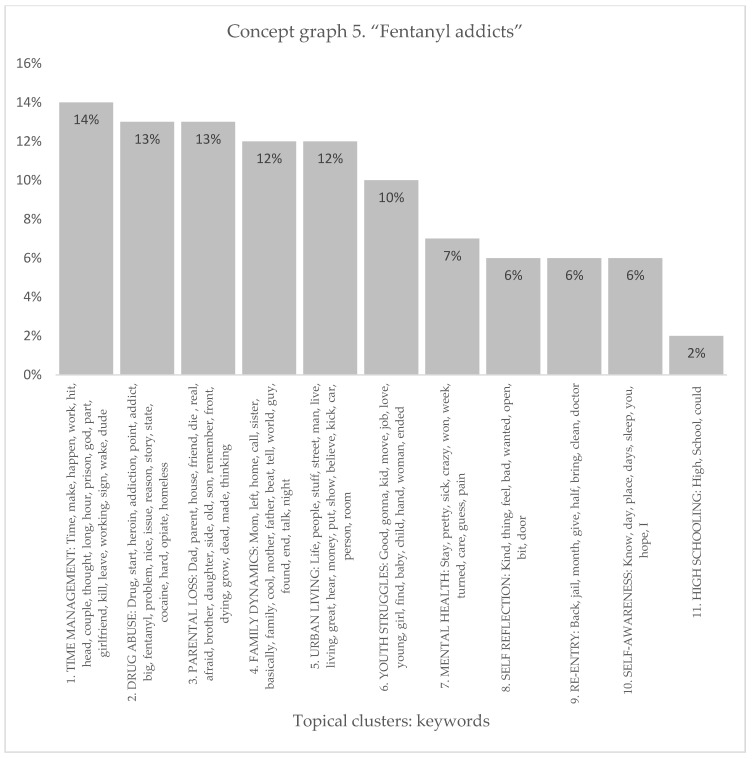
Topical clusters in concept graph 5 “Fentanyl addicts” sample.

**Figure 11 healthcare-11-02439-f011:**
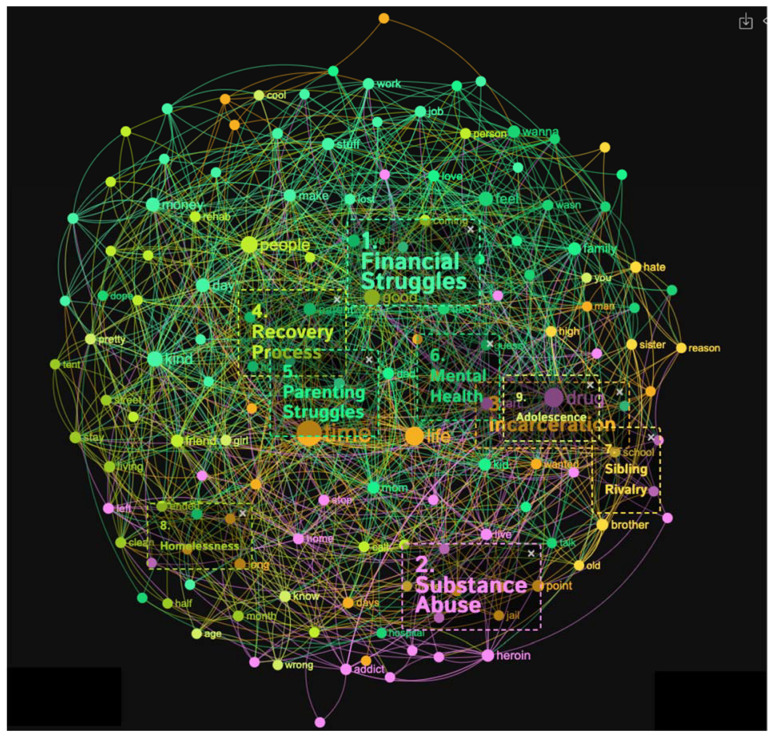
Concept graph 6 “Heroin addicts” sample.

**Figure 12 healthcare-11-02439-f012:**
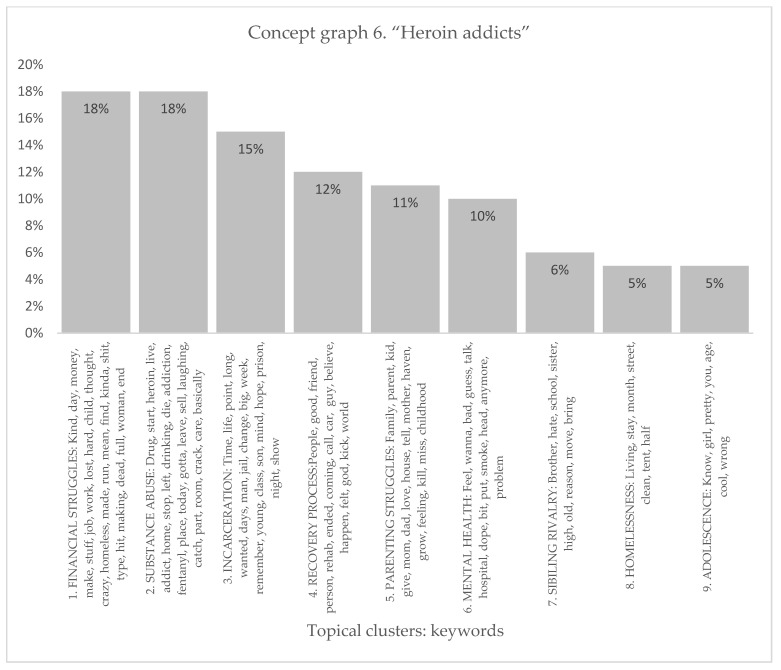
Topical clusters in Concept graph 6 “Heroin addicts” sample.

**Table 1 healthcare-11-02439-t001:** Relevancy assessment and inclusion and exclusion process.

Video Content Relevancy Assessment	Inclusion Criteria	Exclusion Criteria
The video must be any of the interviews from Soft White Underbelly YouTube channel featuring people with a substance-use disorder or recovering people with a substance-use disorder. The videos on the channel must be labeled as drug addicts (type of drug not specifically mentioned), crack addicts, crystal meth addicts, fentanyl addicts, heroin addicts, poly drug addicts, ex-drug addicts, or recovering drug addicts.	Videos with narrative nature of life experience and elaborated description of one’s social reality Videos with coherent dialogue of the interviewer and the interviewee. Videos with English captions	Videos with no narration to one or more life experiences and elaborated description of one’s social reality Videos with no coherent dialogue between the interviewer and the interviewee. Videos with no English caption options

**Table 2 healthcare-11-02439-t002:** Data sample.

Case Identifier	No. of Individuals Interviewed	No. of Videos	No. of Words in Transcript
Drug addicts (Drug type unspecified)	19	20	74,570
“A life on drugs”	6	6	35,860
Crack addicts	50	55	212,133
Crystal meth addicts	58	63	202,251
Fentanyl addicts	78	88	339,028
Heroin addicts	77	83	239,686
**Total**	288	315	1,103,528

**Table 3 healthcare-11-02439-t003:** Patterns of similar themes within conceptual graphs.

Conceptual Graphs	Similar Themes within Other Conceptual Graphs	Organized Phrases According to Themes	Evolution of Themes
Drug addicts		Opioid Crisis, Drug Network, Substance Abuse, Drug AbuseSobriety Journey, Recovery Treatment, Addiction Kick, Addiction Care, Recovery ProcessFamily Dynamics, Family Ties, Parenting, Parental Loss, Parenting Struggles, Sibling RivalryMoney Struggles, Financial StrugglesYouth Culture, High Schooling, High School, Youth StrugglesUrban Life, Urban LivingFamily Moving, Homelessness, Homelessness BeatDrug Metering, location-specific addiction experience, Street Selling, Street Life, Skid RowBehavior Drawing, Self-Care, Self-Reflection, Self-AwarenessTime Cutting, Time Living, Time ManagementEmotions, Reality Struggles/Profanity, Trauma recovery, Mental HealthFamily Loss, Life ChangesRe-Entry, IncarcerationAdolescence	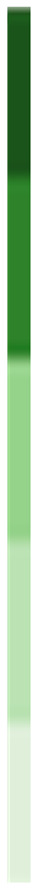
Life on Drug	
Crack addicts	
Crystal meth addicts	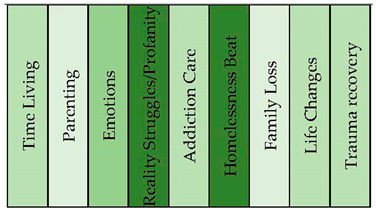
Fentanyl addicts	
Heroin addicts	

## Data Availability

The data presented in this study are available on request from the corresponding author.

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
