# Peer review of "When Unstructured Big Text Corpus Meets Text Network Analysis: Social Reality Conceptualization and Visualization Graph of Big Interview Data of Heavy Drug Addicts of Skid Row"

_healthcare, 2023, doi:10.3390/healthcare11172439_

Round 1

Reviewer 1 Report

This paper introduces a novel methodological approach and analysis tool that can aid health informatics in understanding the social reality of heavy drug addicts in Skid Row, Los Angeles. However, I have some of the comments as follows:

- Title: It is long and ask for reducing it.
- Keywords: It has more than five keywords.
- The authors said that they propose a novel methodological approach.    Clarify it.
- What is significant about your work compared to others? Add the  contributions of this paper in the introduction section.
- The paper has a lot of paragraphs and it  is not organized. The structure of the paper must be organized in a better way. It should include introduction, literature review, materials, methods, results and conclusion and future work if any.
- The abbreviation of SVG in line 46 is not known. fix it and check for others.
- Add Section 6. Limitations of the Study as future work in Section 5. Conclusion.
- Check the format of the paper including the references and make the proper corrections where necessary.

I don't have any potential conflict of interest with regard to this paper, I haven't detected plagiarism (15%) and I don't have any other ethical concerns about this study.

Reviewer 2 Report

1. The introduction is too long, for example, the introduction of GPT-3 and InfraNodus software, which is unnecessary. It is recommended to streamline the introduction to highlight the key points.

2. Some subsection titles are not academic, such as subsection 2.The title of the 2 subsection "Background on the Source of Data, Data Set Selection, and Tool of Analysis" is suggested to be changed to "Materials and methods"; the title of subsection 2.6 "How InfraNodus Tool Perform Text Network Analysis" is suggested to be changed to "Text Network Analysis with using InfraNodus Tool".

3. Table 2 is unnecessary. Its contents are all user guides for the software. You only need to provide the flowchart of the process you used.

4. Subsections 3.1 and 3.2 have a lot of repetitive descriptions about the results. It is suggested to merge and streamline them to highlight the research findings.

Round 2

Reviewer 1 Report

The authors have addressed my comments but they don't provide the clean paper after changes. 

Reviewer 2 Report

Accept in present form